# Pimavanserin and Parkinson’s Disease Psychosis: A Narrative Review

**DOI:** 10.3390/brainsci12101286

**Published:** 2022-09-23

**Authors:** Jamir Pitton Rissardo, Ícaro Durante, Idan Sharon, Ana Letícia Fornari Caprara

**Affiliations:** 1Medicine Department, Federal University of Santa Maria, Santa Maria 97105-900, Brazil; 2Department of Medicine, Federal University of Fronteira Sul, Passo Fundo 99010-121, Brazil; 3NewYork-Presbyterian Brooklyn Methodist Hospital, New York, NY 11215, USA

**Keywords:** pimavanserin, Nuplazid, ACP-103, Parkinson’s disease, psychosis

## Abstract

Pimavanserin (PMV) is the first approved drug for treating hallucinations and delusions in Parkinson’s disease (PD) psychosis. Psychosis is one of the leading causes of nursing home placement in people with PD. Furthermore, hallucinations are a more frequent cause of institutionalization than motor disability or dementia related to PD. The management of PD psychosis involves antipsychotic medications. Most of the drugs in this class directly block dopamine D2 receptors, leading to significantly worsening motor symptoms in patients with PD. The most commonly used medications for managing PD psychosis are quetiapine, clozapine, and PMV. This literature review aims to study pimavanserin’s history, mechanism, clinical trials, and post-marketing experience. PMV is a potent 5-HT2A receptor antagonist/inverse agonist. Moreover, this drug can interact with 5-HT2C receptors. We calculated some physicochemical descriptors and pharmacokinetic properties of PMV. Eight clinical trials of PMV and PD psychosis are registered on ClinicalTrials.gov. Only four of them have complete results already published. Meta-analytic results showed that PMV efficacy is inferior to clozapine. However, PMV has a significantly lower number of side-effects for managing psychosis in PD. Medicare database assessment revealed 35% lower mortality with PMV compared to other atypical antipsychotics. Moreover, sensitive statistical analysis demonstrated that PMV is a protective factor for the risk of falls in individuals with PD.

## 1. Introduction

Psychosis is one of the leading causes of nursing home placement for people with Parkinson’s disease (PD) [1]. In this context, hallucinations are a more frequent cause of nursing home admission than motor disability or dementia related to PD [2]. It affects not only the patient but also the caregiver, marking a crucial stage in PD progression, with severe and deleterious effects on the quality of life of the patient–caregiver relationship [1,3]. Psychosis is a common clinical manifestation in neuropsychiatric conditions leading to dementia, and its prevalence increases with the progression of the underlying disease. Approximately half of the individuals with PD will develop psychosis, hallucinations, or delusions. Therefore, overall psychosis prevalence is more common in PD than in Alzheimer’s disease, vascular dementia, and frontotemporal dementia [4].

The prevalence of PD psychosis broadly varies among studies ranging from 16% to 75% [5]. This difference between the studies can be attributed to inconsistency in diagnosing PD psychosis [6]. In this way, the workgroup of the National Institutes of Neurological Disorders and Stroke and the National Institutes of Mental Health proposed diagnostic criteria for PD psychosis (Table 1) [7]. It is worth mentioning that PD psychosis is a disease of exclusion. Thus, other potential causes of acute psychosis such as stroke, meningitis, electrolyte imbalance, psychoactive medications, and neoplasia should be investigated [8].

PD psychosis is different from other types of psychiatric psychosis. The hallucinations related to PD are mainly visual, which can occur with other minor hallucinations such as a false sense of presence, a false sense of a passage, and illusions [9]. Interestingly, delusions are usually related to spouse infidelity, but they are less prevalent than hallucinations [10]. Some subjects can develop paranoia, but it does not present with unusual thoughts content as seen in individuals with schizophrenia [11].

The underlying mechanism for PD psychosis is poorly understood. It was first assumed that only dopamine influenced the development of psychosis due to small observational studies of this condition, taking into account changes in the levodopa dosage regimen [12]. Recently, studies in animal models of PD psychosis revealed upregulation of serotonin in the postsynaptic membrane of the raphe nucleus. This hyperactivity of serotonin can lead to hyperactivity of dopaminergic pathways in the mesolimbic system [13]. Furthermore, glutamate appears to have a role in the control of this circuit [14].

In some individuals, the management of PD psychosis should involve antipsychotic drugs. However, most of the medications in this class directly block dopamine D2 receptors, leading to significantly worsening motor symptoms in patients with PD. Hence, treatment should be related mainly to atypical antipsychotics primarily involved in the serotonin pathway (Table 2) [15,16,17]. Quetiapine is frequently used off-label for managing PD psychosis. However, there is no evidence of quetiapine efficacy in PD psychosis. Quetiapine minimizes psychotic agitation and distress without modifying psychosis [18]. Clozapine showed efficacy in the management of psychosis related to PD. Nonetheless, clozapine is associated with significant side = effects that require specialized monitoring [19].

This literature review aims to study pimavanserin’s history, mechanism, clinical trials, and post-marketing experience. For further description of this study’s methodology, refer to the Appendix A.

## 2. Pharmacological History of Pimavanserin

In the 1990s, scientists from ACADIA started a chemical genomics program to understand targets for drugs acting on the central nervous system. They observed that some atypical antipsychotics such as clozapine were efficacious for treating psychosis in patients with PD. Thus, they hypothesized that selective 5-HT2A inverse agonism activity might be an appropriate target mechanism for PD psychosis. ACADIA scientists used the Receptor Selection and Amplification Technology^TM^ (R-SAT^TM^) platform to develop efficacious inverse 5-HT2A agonists [20].

Studies with animal models revealed that AC-90179 inhibited MK-801-induced but not amphetamine-induced locomotor activity. Moreover, AC-90179 dose-dependently improved head twitches induced by DOI, a well-known behavior associated with 5-HT2A receptor stimulation [21]. After molecular changes to increase oral bioavailability, PMV (ACP-103) was developed [22]. The structural characteristics and pharmacological selectivity of PMV differentiate it from typical and atypical antipsychotic medications. A rodent model of PD psychosis was used to assess PMV efficacy in PD psychosis treatment. PMV reversed psychosis-like behaviors without augmenting motor problems or blocking the ability of levodopa to improve motor behavior [23].

In the late 2000s, pharmacological studies showed the safety and tolerability of PMV in healthy individuals [24]. About 2 years later, the first clinical trials demonstrated the efficacy of PMV for managing PD psychosis [25].

The first dosage formulation developed for PMV was designed to be liquid. However, the unit dose was changed to solid due to the aesthetically unpleasant taste of the drug ingredients. Moreover, liquid formulations are not well accepted in clinical practice by patients with PD due to possible difficulty swallowing liquids. In this way, dysphagia affects up to 13% of persons 65 years and older and 51% of older persons in nursing homes [26]. Interestingly, a recent study revealed that PMV mixed with certain foods received favorable ratings on measures of palatability and swallowability [27].

On 2 September 2014, ACADIA Pharmaceuticals Inc. received Food and Drug Administration (FDA) breakthrough therapy designation for Nuplazid^®^ (pimavanserin) for PD psychosis. Before that moment, there were no approved medications in the United States for treating PD psychosis. Thus, the US FDA granted breakthrough therapy status to PMV [28].

On 29 April 2016, PMV was approved by the FDA for treating hallucinations and delusions associated with PD psychosis. The nonbinding advisory panel recommendation of 12-to-2 in support of approval that preceded the FDA approval action noted that the drug met a critical need [29]. The approval was based on one multicentric, 6 week, randomized, placebo-controlled, parallel-group study. It is worth mentioning that the scale used to assess efficacy was the Scale for the Assessment of Positive Symptoms PD (SAPS-PD), which is different from the SAPS hallucinations and delusions subscales (SAPS-H+D). In the first studies with SAPS-H+D, the therapeutical efficacy was not as significant as in the SAPS-PD. Moreover, the pivotal trial did not include individuals with dementia and PD psychosis [16].

On 29 June 2018, the FDA approved 34 mg capsules and 10 mg tablet forms of PMV [30]. Previously, users were required to take two 17 mg tablets to achieve the recommended 34 mg daily dose. Furthermore, the lower PMV dose (10 mg) is indicated for patients taking CYP3A4 inhibitors, increasing the therapeutical use in clinical practice.

## 3. Mechanism of Action of Pimavanserin

PMV is mainly considered a potent 5-HT2A receptor antagonist/inverse agonist. Additionally, PMV can interact at 5-HT2C with approximately 40-fold less potency. Inverse agonists have the opposite effect of agonists on intrinsic activity in contrast to blockers or antagonists, which block the effect of agonists but have no inherent impact of their own. Thus, inverse agonists block the actions of agonists at the receptor the same as antagonists and possibly decrease the intrinsic activity that the receptor has in the absence of its agonist (Figure 1) [31].

Serotonin, 5-HT, and 5-hydroxytryptamine receptors are a group of G protein-coupled receptors and ligand-gated ion channels found in the central and peripheral nervous systems. The 5-HT2A receptor is a subtype of the 5-HT2 receptor, a cell surface receptor with several intracellular locations. 5-HT2A is believed to be the most excitatory receptor subtype among the serotonin receptors [32]. Interestingly, the clinical significance of this receptor was first noticed in serotoninergic psychedelic drugs such as lysergic acid diethylamide [33].

The relationship between serotonin receptors and psychosis, hallucinations, or delusions is poorly understood. However, recent studies showed that some neuropsychiatric diseases such as schizophrenia and addiction present abnormalities in the 5-HT2A receptor gene [34]. In addition to sensorimotor effects, this receptor is also associated with pain perception, motivation, and emotional regulation [35].

The pharmacological importance of the PMV interaction with 5-HT2C is scarce. It was observed that high doses of PMV lead to the recruitment of 5-HT2C receptors after the saturation of 5-HT2A receptors [22]. Noteworthily, the 5-HT2C receptor is related to generalized anxiety disorder, obsessive–compulsive disorder, attention-deficit/hyperactive disorder, and depressive disorder [35].

We calculated the chemical and pharmacological properties of PMV using the SwissADME tool (Figure 2) [36]. These properties help identify compounds suitable for oral use. All the parameters analyzed were within the normal range, except for slightly high flexibility related to the binding properties of the receptor. Refer to the Appendix A for a complete description of physicochemical descriptors and pharmacokinetic characteristics of PMV (Appendix A) [20,36].

## 4. Pimavanserin Clinical Trials

The results of the clinical trials related to the PMV indicated for PD psychosis are summarized in Table 3 [16,25]. A table was uploaded in the Appendix A with all the clinical trials related to pimavanserin registered on ClinicalTrials.gov (Appendix A). Moreover, a figure is provided with the conditions assessed in these clinical trials (Figure 3) [37,38,39,40,41,42]. Refer to the Appendix A for the mechanism proposed by 5-HT2A and schizophrenia, Tourette syndrome, depressive disorder, impulsivity, dyskinesia, and insomnia (Appendix A).

Eight clinical trials of PMV and PD psychosis are registered: NCT02762591, NCT00477672, NCT00658567, NCT01174004, NCT01518309, NCT00550238, NCT04292223, and NCT04373317. The total number of enrolled people with PD psychosis is estimated at 1529. However, we only have complete results already published in four studies. NCT02762591 was registered to provide patients with PD psychosis access to pimavanserin until the product received marketing approval from the US FDA and is commercially available.

The primary outcome of the clinical trials involved two types of SAPS scales. NCT00658567, NCT00477672, and Meltzer et al. assessed SAPS-H+D. On the other hand, NCT01174004 assessed SAPS-PD. The secondary outcome of the clinical trials was the influence of PMV on motor symptoms, which was investigated with the Unified Parkinson’s Disease Rating Scale (UPDRS) Part II (Activities of Daily Living) and Part III (Motor Examination).

Meltzer et al. also studied other scales such as Parkinson’s Psychosis Rating Scale (PPRS), Clinical Global Impression—Severity (CGI-S), and Epworth Sleepiness Scale (ESS). Cummings et al. assessed Mini-Mental Status Examination (MMSE), Neuropsychiatric Inventory (NPI), Clinical Global Impression—Severity (CGI-S), and Caregiver burden scale (CBS). Refer to the Appendix A for a complete description of the scales used for the clinical trials related to PMV and PD psychosis (Appendix A) [43,44,45,46,47,48,49,50,51,52].

## 5. Pimavanserin Clinical Experience

### 5.1. Pimavanserin Efficacy

The approval of PMV by the FDA with only one “pivotal trial” and modified scales for psychosis assessment left some experts in doubt about the effectiveness of PMV [16]. However, the clinical trial for approval was not the only study that showed efficacy. Two other clinical trials developed simultaneously showed significant results for PMV in PD psychosis management. Moreover, the changes in scales were probably needed due to their low specificity for PD symptoms and broad range scoring system, which went from zero to 20 items. Furthermore, SAPS-H+D was aimed at assessing psychotic symptoms of schizophrenia [49]. Additionally, this scale assesses various types of hallucinations and delusions that are not commonly associated with PD [47]. The new SAPS-PD scale ranges from zero to nine items and is specifically designed for psychosis associated with PD [50].

Mansuri et al. aimed to evaluate the safety and efficacy of PMV in treating PD psychosis on the basis of data from four clinical trials with complete published results [53]. They observed that PMV was associated with a significant reduction in psychosis (SAPS, mean difference: –1.55 [–2.71, –0.379], *p* = 0.009). The groups had similar composite scores for motor symptoms. Interestingly, PMV was protective against orthostatic hypotension (risk ratio: 0.33 [0.30, 0.37], *p* < 0.001).

Currently, no clinical trials have compared atypical antipsychotics and PMV to manage PD psychosis. In this way, a network meta-analysis between clozapine and PMV was performed to assess the efficacy and safety of PMV compared to atypical antipsychotics for psychosis in PD [54]. The study involved 17 clinical trials and showed that clozapine is efficacious with a low impact on motor functions. An important fact is that PMV efficacy is inferior to clozapine. Notwithstanding, PMV has a favorable profile for treating psychosis in PD.

Isaacson et al. evaluated the efficacy and tolerability of PMV in an open-label extension study for a more extended period [55]. They observed that the beneficial effects of PMV in the 6 week core study were maintained, supporting the durability of response to PMV. Sellers et al. showed similar results with clinical improvement in psychosis in 76% of the individuals, which was maintained for a long-term follow-up of 2 years [56]. Moreover, cognitive impairment and deep brain stimulation do not alter the effect of PMV. However, subjects with earlier PD psychosis onset may respond better to PMV [57].

### 5.2. Switching from Off-Label Antipsychotics to Pimavanserin

Some patients might require switching from off-label antipsychotics to PMV due to inefficacy in improving psychotic symptoms or emergence of severe side-effects. Black et al. published a consensus recommending adding PMV 34 mg daily to the continuous quetiapine dose for 4 weeks and continuing clozapine for 6 weeks, followed by tapering antipsychotics [58]. High-dose quetiapine (>100 mg/day) should be reduced by 50% weekly until reaching 12.5 mg/day and then withdrawn. Low-dose quetiapine (<100 mg/day) should be reduced by 25% weekly until reaching 12.5 mg/day and then removed. High-dose clozapine (>100 mg/day) should be reduced by 25 mg weekly until discontinued. Low-dose clozapine (<100 mg/day) should be reduced by 6.25 mg weekly until discontinued. While tapering the antipsychotics, if the patient has an emergence of psychosis, the prescriber can return to the previous dose level, and the tapering should be attempted again after 1 week.

### 5.3. Post-Marketing Surveillance and Experience

Concerns about PMV safety have been expressed by consumer watchdogs such as the Institute for Safe Medical Practices and the popular press [59]. Moreover, a study assessing the Medicare database showed that patients taking PMV for PD psychosis, compared to those with no treatment over 1 year, had an increased risk for death [60]. Nevertheless, another study of the Medicare database revealed that mortality was about 35% lower with PMV during the first 6 month follow-up period than with atypical antipsychotics for managing PD psychosis [61].

Brown et al. retrospective analyzed the adverse event case reports submitted to the FDA’s Adverse Event Reporting System (FAERS) from 2016 through 2019 [62]. They aimed to compare PMV with treatment alternatives and other atypical antipsychotics within a population of people with PD psychosis. Their results showed that PMV was not associated with excess reports of death. Additionally, Brown et al. noticed a substantial bias related to reports of PMV adverse events in 2018, probably associated with consumer safety groups and media reports, raising concerns based on the absolute numbers of death reports in individuals receiving PMV.

Ballard et al. performed a multiyear, open-label study assessing the long-term safety and tolerability of PMV [63]. Long-term treatment with pimavanserin 34 mg once daily demonstrated a favorable benefit/risk profile with no unexpected safety concerns. Mortality rates suggested no increased risk following long-term therapy. The study was performed for 11 years and revealed that the most common side-effects were falls, urinary tract infections, and hallucinations.

A recent cohort study after sensitive statistical analysis demonstrated a decreased risk of falls and fractures in patients with PD psychosis treated with PMV versus atypical antipsychotics [64]. The incidence ratio estimated was 0.55 (95% CI 0.34–0.86). Therefore, additional research is warranted when PMV is more widely used among patients with PD psychosis.

### 5.4. Pimavanserin Clinical Experience

PMV, when available, should be the first-line therapy for managing PD psychosis. Furthermore, this drug should be started at symptom onset since PD psychosis progression is unpredictable. Atypical antipsychotics are recommended as a second choice for PMV when PD psychosis is not sufficiently controlled. Low-dose quetiapine may be a second-line agent in an initial trial [15].

Dashtipour et al. reported a case series of PMV treatment for PD psychosis with clinical tips. Table 4 was adapted from the study of Dashtipour et al. [65]. It is worth mentioning that US prescribers of PMV must provide medical documentation of psychosis and the reason for its indication, which can sometimes affect the use of PMV for behavioral management [66].

In cases of PMV-resistant psychosis, one possible approach is PMV tapering and clozapine titration. Lake et al. reported an elderly female with PD psychosis resistant to quetiapine and PMV monotherapy regimen. A trial of clozapine after a suboptimal response to PMV led to further symptom improvement [67]. Noteworthily, the decision should have a patient-centered approach due to the high potential side-effect burden.

PMV was first approved with the recommended swallowing of the entire capsule or tablet. Recent studies revealed that food does not affect PMV bioavailability. Moreover, PMV can be given orally by emptying the capsule contents into soft foods or liquids. Some examples of the tested products in which PMV capsules were opened and the contents sprinkled are applesauce, orange juice, and water [68].

Another important clinical topic is the possible interactions of PMV with medicines used in the pharmacotherapy of patients with PD. PMV could have a synergistic effect on the prolongation of QT when combined with other drugs that affect QT. However, Bugarski-Kirola et al. showed that this theoretical adjunctive PMV effect with background antipsychotic treatments is not clinically significant [69].

## 6. Future Directions

Future studies of PD psychosis causes and pathophysiology are mandatory for developing targeting therapy. In this context, investigations of PMV and PD psychosis with larger samples and diverse backgrounds should be done. Noteworthily, more than 90% of the patients included in the clinical trials with PMV were white.

Head-to-head clinical trials comparing efficacy, side-effects, and costs regarding quetiapine, clozapine, and PMV in managing PD psychosis are mandatory. This information will significantly impact current guidelines and clinical practice of PD therapy.

Another fact that should be clarified is the long-term outcomes of PMV use. There were no clinical trials that assessed the long-term follow-up of PMV. However, cross-sectional and cohort studies showed maintenance of the improvement in PD psychosis with PMV. Moreover, further studies are mandatory for the duration of the treatment since some individuals could benefit from short-term therapies.

## 7. Conclusions

In sum, PMV for managing PD psychosis is efficacious and well tolerable. The most frequent side-effects of PMV are mild to moderate and usually related to gastrointestinal symptoms. Currently, PMV use is safe and related to significant improvement in the patient’s quality of life regarding sleep and caregiver–patient relationship. Further studies evaluating a large, broad population of PD psychosis are mandatory. Moreover, clinical trials of PD psychosis comparing quetiapine, clozapine, and PMV should be performed.

## Figures and Tables

**Figure 1 brainsci-12-01286-f001:**
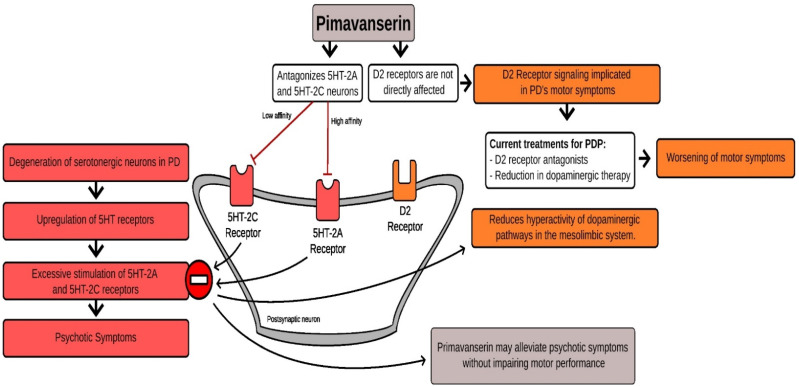
Hypothetical mechanism of action of pimavanserin in Parkinson’s disease psychosis.

**Figure 2 brainsci-12-01286-f002:**
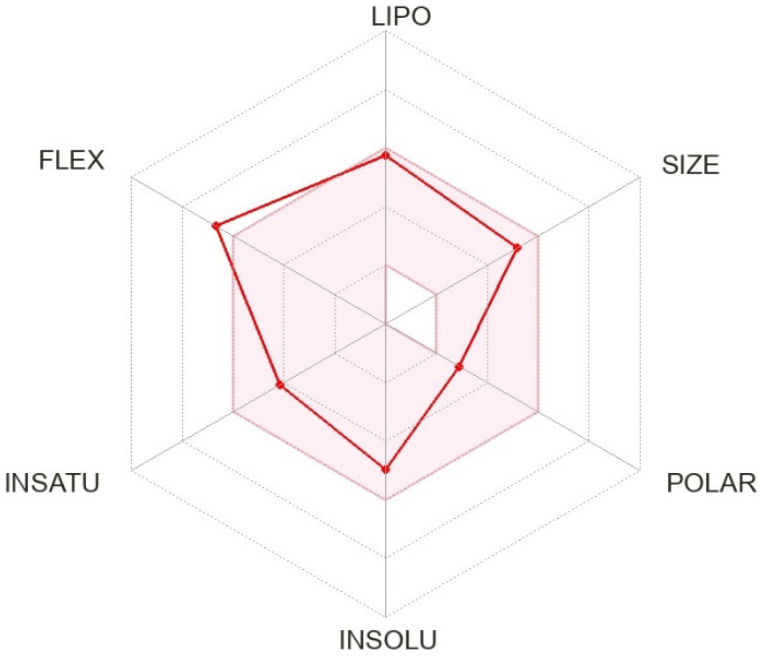
Physicochemical properties of pimavanserin (ACP-103; BVF-048).

**Figure 3 brainsci-12-01286-f003:**
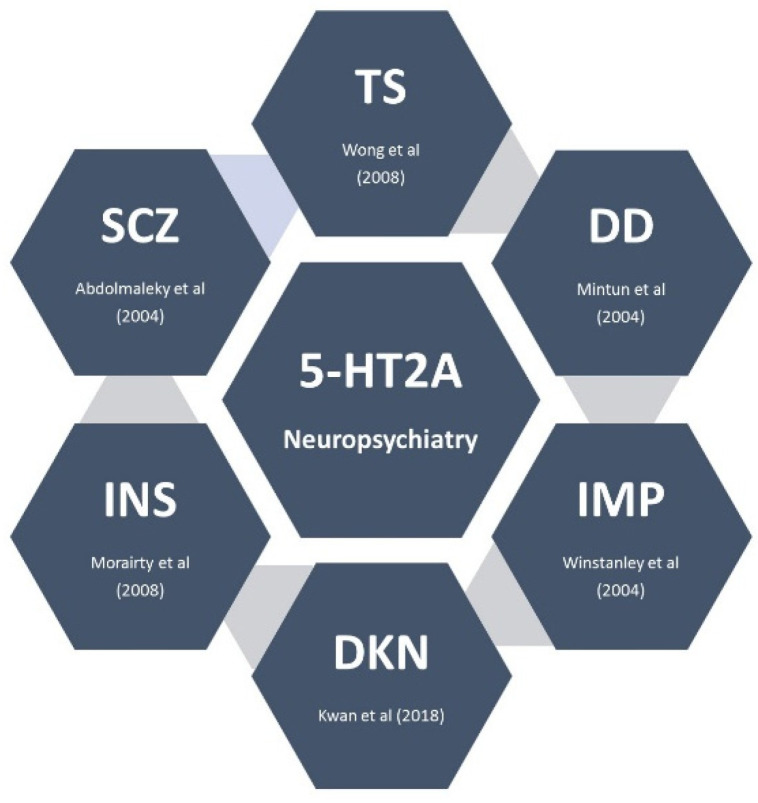
Neuropsychiatric disorders and 5-HT2A receptor. This figure summarizes the conditions studied in the clinical trials with pimavanserin. Abbreviations: IMP: impulsivity; INS: insomnia; DD: depressive disorder; DKN: dyskinesia; SCZ: schizophrenia; TS: Tourette syndrome [37,38,39,40,41,42].

**Table 1 brainsci-12-01286-t001:** NINDS and NIMH work group diagnostic criteria for PD psychosis.

The presence of at least one of the following symptoms of psychosis: (a) Illusions (b) False sense of presence (c) Hallucinations (d) Delusions	1. A primary diagnosis of PD
	2. Symptoms of psychosis occur after the onset of PD
	3. The duration of symptoms of psychosis are recurrent or continuous for at least one month
	4. Exclusion of alternative diagnosis

NINDS: National Institutes of Neurological Disorders and Stroke; NIMH: National Institutes of Mental Health; PD: Parkinson’s disease; UKPDS: United Kingdom Parkinson Disease Society [7].

**Table 2 brainsci-12-01286-t002:** American Academy of Neurology classification of recommendations for treatment of Parkinson’s disease psychosis.

Medication	Current Evidence
Pimavanserin	Level B (should be considered)
Clozapine	Level B (should be considered)
Quetiapine	Level C (may be considered)

Source: [15,16,17].

**Table 3 brainsci-12-01286-t003:** Clinical trials of pimavanserin in the management of Parkinson’s disease psychosis.

Study	NCT00658567			NCT00477672			Meltzer et al.		Cumming et al. (NCT01174004)	
Year	2009			2009			2010		2014	
Type of study	Randomized parallel assignment, quadruple masking			Randomized parallel assignment, quadruple masking			Multicenter, randomized, placebo-controlled, double-blind		Randomized parallel assignment, quadruple masking	
Total number of participants	123			298			60		199	
Intervention	Pimavanserin 10–20 mg and placebo			Pimavanserin 10–40 mg and placebo			Pimavanserin 20–60 mg		Pimavanserin 40 mg and placebo	
Primary outcome	Change in SAPS score from baseline to day 42			Change in SAPS score from baseline to day 42			Change in SAPS score from baseline to day 28		Change in SAPS score from baseline to day 43	
Secondary outcome	Change in UPDRS II/III score from baseline to day 42			Change in UPDRS II/III score from baseline to day 42			Change in UPDRS II/III score from baseline to day 28		Change in UPDRS II/III score from baseline to day 43	
Groups	PMV10	PMV20	Pla	PMV10	PMV40	Pla	PMV	Pla	PMV	Pla
N	41	41	39	99	98	98	29	31	95	90
Age mean (SD)	71 (7.4)	72.1 (8.2)	73 (7.9)	69.0 (8.6)	69.4 (7.8)	69.6 (9.7)	72.3 (1.4)	69.6 (1.6)	72.4 (6.6)	72.4 (7.9)
Sex (male)	26	24	27	63	74	51	26	20	64	52
Race (white)	-	-	-	-	-	-	28	31	90	85
Primary endpoint	-	−6.5	−4.4	−5.8	−6.7	−5.9	−1.9	−0.2	−5.7	−2.7
Secondary endpoint	-	−3.9	−1.8	−1.4	−3.1	−2.9	−3.0	−3.8	−1.6	−1.4

SAPS: Scale for the Assessment of Positive Symptoms. UPDRS: Unified Parkinson’s Disease Rating Scale (UPDRS) Part II (Activities of Daily Living) and Part III (Motor Examination) PMV: pimavanserin; Pla: placebo. NCT00658567 and NCT00477672 data extracted from ClinicalTrials.gov.

**Table 4 brainsci-12-01286-t004:** Pimavanserin clinical experience by Dashtipour et al. adapted by Rissardo et al.

1. Low PMV dose (17 mg/day; 10 mg/day) may be effective in some patients
2. PMV may be added without disrupting or adversely affecting other multidrug PD regimens
3. PMV may be effective for managing PD psychosis in individuals with deep brain stimulation
4. PMV and another antipsychotic may be necessary to manage PD psychosis
5. After controlling PD psychosis symptoms with PMV, clinicians can increase the dose of dopaminergic drugs for better motor control

PD: Parkinson’s disease; PMV: pimavanserin. Source: [65].

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
