# Peer review of "Pimavanserin and Parkinson’s Disease Psychosis: A Narrative Review"

_brainsci, 2022, doi:10.3390/brainsci12101286_

Round 1

Reviewer 1 Report

Parkinson’s disease is the second most common neurodegenerative disease affecting more and more people. Only symptomatic treatments are available, nevertheless, treatments of non-motor symptoms such as hallucinations is of great importance. Therefore this review on pimavanserin (PMV) is timely and of significance.

However, there are minor comments/questions concerning the manuscript.

In the cases of the completed clinical trials (table 3), are there data whether the patients were treated with levodopa or other drugs at the same time?

For the sake of general readers, describe the similarities and differences of SAPS-H+D and SAPS-PD in more detail.

Please, pay attention to the abbreviations. For example, 5-HT2A is mentioned in the abstract and first in line 85, but its description can be found in line 137. Figure 2 is also difficult to understand.

Author Response

Parkinson’s disease is the second most common neurodegenerative disease affecting more and more people. Only symptomatic treatments are available, nevertheless, treatments of non-motor symptoms such as hallucinations is of great importance. Therefore this review on pimavanserin (PMV) is timely and of significance.

However, there are minor comments/questions concerning the manuscript.

In the cases of the completed clinical trials (table 3), are there data whether the patients were treated with levodopa or other drugs at the same time?

Dear Reviewer, we would like to highlight that this is an interesting question that we already pointed out in other reviews regarding movement disorders associated with drugs. We attempt to search again this information in the clinical trials (NCT00658567, NCT00477672, Meltzer et al., and NCT01174004). There is no information in the “study results” section regarding other medications. We believe that the authors of those manuscripts performed a similar baseline group, but no data related to the use of anti-parkinsonian drugs is done. It is worth mentioning that this is a common concern in studies involving Parkinson’s disease. The most common approach in the studies is standardized the data with the years of the disease.

For the sake of general readers, describe the similarities and differences of SAPS-H+D and SAPS-PD in more detail.

We believe that this question can be addressed with supplementary material. Supplementary Material 5 describes the scales performed on pimavanserin studies of Parkinson’s disease psychosis. The number of items, purpose, and considerations are provided regarding these scales. The scales assessed were CGI-S, CGI-I, ESS, MMSE, NPI, PPRS, SAPS-H+D, SAPS-PD, UPDRS II/III, and CBS.

Please, pay attention to the abbreviations. For example, 5-HT2A is mentioned in the abstract and first in line 85, but its description can be found in line 137. Figure 2 is also difficult to understand.

Dear Reviewer, we believe that including the full description of this term will not affect the understanding of the manuscript. 5-HT2A is a standardized description of serotonin receptors. To support this opinion, we will show some article titles that can be found on PubMed below. Regarding Figure 2, the complexity of the 5-HT2A receptor and the neuropsychiatric pathologies involved are difficult to understand. This figure summarizes the hypothesis of approximately thirty clinical trials related to developing new targets for managing neurologic diseases.

McFarland, K.; Price, D.L.; Bonhaus, D.W. Pimavanserin, a 5-HT2A inverse agonist, reverses psychosis-like behaviors in a rodent model of Parkinson’s disease. Behavioural pharmacology 2011, 22, 681-692, doi:10.1097/FBP.0b013e32834aff98.

Stahl, S.M. Mechanism of action of pimavanserin in Parkinson’s disease psychosis: targeting serotonin 5HT2A and 5HT2C receptors. CNS spectrums 2016, 21, 271-275, doi:10.1017/s1092852916000407.

Schmidt, C.J.; Sorensen, S.M.; Kehne, J.H.; Carr, A.A.; Palfreyman, M.G. The role of 5-HT2A receptors in antipsychotic activity. Life sciences 1995, 56, 2209-2222, doi:10.1016/0024-3205(95)00210-w.

Winstanley, C.A.; Theobald, D.E.; Dalley, J.W.; Glennon, J.C.; Robbins, T.W. 5-HT2A and 5-HT2C receptor antagonists have opposing effects on a measure of impulsivity: interactions with global 5-HT depletion. Psychopharmacology 2004, 176, 376-385, doi:10.1007/s00213-004-1884-9.

Morairty, S.R.; Hedley, L.; Flores, J.; Martin, R.; Kilduff, T.S. Selective 5HT2A and 5HT6 receptor antagonists promote sleep in rats. Sleep 2008, 31, 34-44, doi:10.1093/sleep/31.1.34.

Mintun, M.A.; Sheline, Y.I.; Moerlein, S.M.; Vlassenko, A.G.; Huang, Y.; Snyder, A.Z. Decreased hippocampal 5-HT2A receptor binding in major depressive disorder: in vivo measurement with [18F]altanserin positron emission tomography. Biological psychiatry 2004, 55, 217-224, doi:10.1016/j.biopsych.2003.08.015.

Reviewer 2 Report

The review by Rissardo et al. on PMV for treating hallucinations and delusions in Parkinson's disease psychosis is generally well written. I suggest improving the readability and clarity of Table 1 alongside the relevant comment in the main text, as well as discussing possible interactions with medicines used in the pharmacotherapy of PD patients.

Author Response

The review by Rissardo et al. on PMV for treating hallucinations and delusions in Parkinson's disease psychosis is generally well written. I suggest improving the readability and clarity of Table 1 alongside the relevant comment in the main text, as well as discussing possible interactions with medicines used in the pharmacotherapy of PD patients.

Dear Reviewer,

We believe that no modification in Table 1 should be done. Table 1 describes the NINDS and NIMH work group diagnostic criteria for Parkinson’s disease psychosis. The table was only slightly modified, but maintain the main characteristics for the diagnosis. To be more specific, only PD that is the abbreviation for Parkinson’s disease was done. Therefore, we do not believe that its modification would improve its quality since the table is standard.

Regarding pharmacotherapy interaction, this is a common concern in Parkinson’s disease studies. The most common approach is to standardize the individuals based in the years of disease of Parkinson’s disease. We agree with the Reviewer that new standardized approaches are mandatory especially for commonly studies neurological diseases such as Parkinson’s disease. Therefore, we do not have a clear description of how much other drugs can influence the response of pimavanserin.

Another important clinical topic is the possible interactions of PMV with medicines used in the pharmacotherapy of patients with PD. PMV could have a synergistic effect on the prolongation of QT when combined with other drugs that affect QT. However, Bugar-ski-Kirola et al. showed that this theoretical adjunctive PMV effect with background anti-psychotic treatments is not clinically significant [69].

Bugarski-Kirola, D.; Nunez, R.; Odetalla, R.; Liu, I.Y.; Turner, M.E. Effects of adjunctive pimavanserin and current antipsychotic treatment on QT interval prolongation in patients with schizophrenia. Front Psychiatry 2022, 13, 892199, doi: 10.3389/fpsyt.2022.892199